# Tuberculosis in the Elderly

**DOI:** 10.3390/jcm10245888

**Published:** 2021-12-15

**Authors:** Pauline Caraux-Paz, Sylvain Diamantis, Benoit de Wazières, Sébastien Gallien

**Affiliations:** 1Service de Maladies Infectieuses et Tropicales, Hôpital Intercommunal de Villeneuve-Saint-Georges, 94190 Villeneuve-Saint-Georges, France; 2Service de Maladies Infectieuses et Tropicales, Hôpital de Melun, 77000 Melun, France; sylvain.diamantis@ghsif.fr; 3Unité de Recherche DYNAMIC, Université Paris-Est Créteil, 94000 Créteil, France; sebastien.gallien@aphp.fr; 4Service de Gériatrie, CHU de Nîmes, 30900 Nîmes, France; benoit.de.wazieres@chu-nimes.fr; 5Service de Maladies Infectieuses, CHU Mondor—APHP, 94000 Créteil, France

**Keywords:** tuberculosis, elderly, aging

## Abstract

The tuberculosis (TB) epidemic is most prevalent in the elderly, and there is a progressive increase in the notification rate with age. Most cases of TB in the elderly are linked to the reactivation of lesions that have remained dormant. The awakening of these lesions is attributable to changes in the immune system related to senescence. The mortality rate from tuberculosis remains higher in elderly patients. Symptoms of active TB are nonspecific and less pronounced in the elderly. Diagnostic difficulties in the elderly are common in many diseases but it is important to use all possible techniques to make a microbiological diagnosis. Recognising frailty to prevent loss of independence is a major challenge in dealing with the therapeutic aspects of elderly patients. Several studies report contrasting data about poorer tolerance of TB drugs in this population. Adherence to antituberculosis treatment is a fundamental issue for the outcome of treatment. Decreased completeness of treatment was shown in older people as well as a higher risk of treatment failure.

## 1. Introduction

Tuberculosis (TB) caused by the bacterium *Mycobacterium tuberculosis* is one of the most frequent infectious diseases in the world and remains a public health problem in terms of diagnosis and treatment. Every year 10 million people fall ill with TB, and despite being a preventable and curable disease 1.5 million people die from TB each year [1].

In the elderly population, several factors coexist to make TB a specific issue. Immunodeficiency related to aging, the potential for added immunodepression conditions related to other aged comorbidities, and potential interactions between antituberculosis drugs and other additional medications. In addition, few data are available specifically about TB in older subjects.

The purpose of this review is to provide a summary, synthesis and critical evaluation of the literature relevant to specific features of epidemiology, pathophysiology, diagnosis, treatment and outcome of tuberculosis in the elderly.

## 2. Epidemiology

### 2.1. TB Is a Common Communicable Disease, including in the Elderly, Although It Is Decreasing

About one-quarter of the world’s population is estimated to be infected by TB bacteria, and most of the people who develop TB disease live in low-income and middle-income countries; however, TB is present all over the world [1]. Geographically, in 2019, most TB cases were in the World Health Organization (WHO) regions of South-East Asia (44%), Africa (25%) and the Western Pacific (18%), with smaller incidence in the Eastern Mediterranean (8.2%), the Americas (2.9%) and Europe (2.5%) [2]. Eight countries in Asia and sub-Saharan Africa accounted for two thirds of the global total: India (26%), Indonesia (8.5%), China (8.4%), the Philippines (6.0%), Pakistan (5.7%), Nigeria (4.4%), Bangladesh (3.6%) and South Africa (3.6%).

TB affects people of both sexes and all age groups, but the highest burden is in adult men, who accounted for 56% of all TB cases in 2019; by comparison, adult women accounted for 32% and children for 12%. The male:female ratio of incident TB cases for all ages ranged from 1.3 in the WHO Eastern Mediterranean Region to 2.1 in the European and Western Pacific regions. Among all TB cases, 8.2% were among people living with HIV.

Concerning TB incidence according to age, the group with the greatest percentage of TB cases has an age range of 25 to 54 years. However, in the WHO regions of the Eastern Mediterranean, South-East Asia and Western Pacific, the TB epidemic is most prevalent in the elderly with a progressive increase in the notification rate with age, and a peak among those aged 65 years or over.

Specifically in France in 2019, the percentage of declared cases of tuberculosis disease was 17.5% in subjects >65 years old, 9.8% in >75 years and 6.3% in >80 years [3]. The second and the third highest incidences of TB cases in 2019 were for people over 65 years old and with greater incidence in women than men, though TB incidence generally is decreasing among people older than 65 years [1].

Most cases of tuberculosis in the elderly are linked to the reactivation of lesions that have remained dormant for several decades. The awakening of these lesions is attributable to changes in the immune system related to senescence, notably the decline in the ability to reactivate previously acquired immunity, and/or additional factors [4,5].

### 2.2. TB Outcome

The global treatment success rate for people newly diagnosed with TB is 85% [2]. Decreased completeness of treatment has been shown in older people as well as a higher risk of treatment failure in people >85 years old [6,7]. In addition, a potentially unfavorable outcome was also significantly associated with TB severity, residing in congregate settings, homelessness, having a smear-positive pulmonary case, being born abroad and residing in France for <2 years, history of previous anti-TB treatment and age >85 years [7].

In terms of mortality, the estimate of the death rate by TB was 14.94 for 100,000 infected people in 2017 including 208,000 (range 177,000–242,000) deaths from TB among HIV-positive people [2,8]. Globally, the annual number of TB deaths is falling worldwide, with 31% and 19% reductions in TB deaths from 2015 to 2019 in the WHO European and the African Regions, respectively. According to age, TB mortality rate remains the highest in older patients, specifically in the 70 years and older age range, but has decreased over time since 1990 as represented on Figure 1 [8].

### 2.3. Drug-Resistant TB

Worldwide, around half a million people have developed rifampicin-resistant (RR) TB, of which 78% had multidrug-resistant (MDR) TB, whereas in India, China and the Russian Federation there is a concentration of 49% drug-resistant TB. Globally in 2019, 3.3% of new TB cases and 17.7% of previously treated cases had MDR/RR-TB. For example, in France, since 2012 the number of notified drug-resistant TB cases has increased overall but it is low with around 80 to 100 cases per year (75 cases in 2020). This increase is largely linked to strains isolated from patients born in Eastern Europe, and no signal involving the geriatric population has been demonstrated. There are no global aggregate data between the prevalence of MDR TB strains in the elderly. For example, drug resistance rates were significantly lower in the elderly for all TB drug resistance (6.5 vs. 13.9%) and for MDR (0.6 vs. 3.1%) in Germany in 2011 [9]. Anyway, because TB in the elderly is mostly due to a reactivation of latent lesions, the risk of TB resistance is very low, in patients born in Western Europe or in North America. 

### 2.4. HIV and TB in the Elderly

As the risk of progression to TB disease is associated with cellular immunosuppression, HIV infection is significantly associated with substantially higher incidence and mortality than in HIV-negative people (WHO). Because older adults living with HIV represent a heterogeneous population equally distributed between people aging with HIV and people who acquired HIV at an older age [10], HIV screening is needed when TB is diagnosed, whatever the patient’s age. As the proportion of elderly people is progressively growing per year faster than the population as a whole, the number of older adults living with HIV is increasing as well [11,12]. In France, 20% of new HIV diagnoses were 50 years old or older in 2016, the vast majority being in the 50–59 years old group (73%). However, 60 to 69 year-olds represented 23% and those aged 70 and over 4% [13]. In addition, older people were more likely to have discovered their HIV status at an advanced stage of infection than 25 to 49-year-olds (38% vs. 26%, *p* < 0.0001) when tuberculosis is a leading cause of death and hospitalization in people living with HIV [14].

Indeed, several factors must be integrated in the management of TB in elderly HIV subjects, such as cotreatment of tuberculosis with antiretroviral therapy and its optimal timing of initiation, drug-drug interactions, drug tolerability, and the prevention and treatment of tuberculosis-associated immune reconstitution syndrome [15].

## 3. Pathophysiology 

### 3.1. Modified Pulmonary Function in the Elderly and Structural Changes in the Senescent Lung Promote Lung Infection

Aging is associated with a progressive decrease in lung function [16]. Because of aging, an individual’s reserve is diminished, but this decrease is heterogeneous between individual subjects. The presentation of respiratory disorders may differ in the elderly, especially because of a lack of perception of symptoms such as dyspnea. 

Many physiological changes are associated with ageing, such as a decrease in the elastic recoil of the lung, a decrease in compliance of the chest wall, and a decrease in the strength of respiratory muscles. Under-nutrition, also common in the elderly, can produce sarcopenia and respiratory muscle dysfunction. In addition, calcification, changes in the shape of the thorax, dorsal kyphosis and increased anteroposterior diameter, has a negative effect on muscles’ force-generating capabilities. Decreased forced expiratory flow rates and lung elastic recoil may also compromise the efficacy of clearance of airway secretions by coughing. Moreover, increased incidence of fluid and/or solid aspiration into the lung with old age, and age associated inflammatory disease such as chronic obstructive pulmonary disease (COPD) and pulmonary fibrosis [17], make the elderly more likely to have a pulmonary environment that favors the establishment of infection, including tuberculosis infection.

### 3.2. Risk Factors of TB in the Elderly Are Linked Simultaneously to Various Parameters

Immunosenescence, individual susceptibility (accumulated co-morbidities, malnutrition, functional dependence), favouring treatments like corticoids or immunosuppressants, communal living for some (life in Care Home for Dependent Elderly People) and closeness that increases contacts between residents and healthcare staff, are specific geriatric settings which favour TB infection.

The risk of infection, especially the risk of serious infection, increases with age and predisposes to reactivation of TB [18], the reasons being that increased susceptibility in elderly people include immunosenescence, various anatomical and physiological changes linked to ageing, as well as malnutrition and comorbidities.

Immunosenescence is consistent in part with lymphopaenia, due to the reduced output of immunologically naïve T cells from the thymus but also impaired ability of reactive T cells to achieve immunological memory, both favoring intracellular pathogen infections such as TB [19]. In addition, immunosenescence is influenced by infections and comorbidities that model immune repertories, previous infections, or chronic infections by latent viruses (herpes virus, especially the cytomegalovirus) [20]. The increase in concentrations of inflammatory cytokines (called “inflammaging”) and, in general, tissue ageing, contribute to high risk of infection [21]. Increased oxidative stress and chronic inflammation, reduced phagocytic capacity of neutrophils and macrophages with a possible link to activity of reactive oxygen species, and impaired activities of natural killer cells resulting from hyperglycaemia, also play important permissive roles for *M. tuberculosis* to survive intracellularly [22,23].

## 4. Diagnosis

### 4.1. Clinical Findings

The clinical presentation of tuberculosis in the elderly has its own specificities, but the usual symptoms are a progressive history, perhaps over several months, of discomfort, weight loss, cough, dyspnea, shivering, fever and night’s sweats, and sometimes pains in the chest [24]. In advanced disease, fibrosis with resultant contraction of the upper lobe produces flattening of the chest, tracheal shift, diminished percussion and altered breath sounds [25]. Pleural involvement with either effusions or thickening is much more common in the elderly, occurring in up to 50% of cases [26]. In a retrospective UK study the less specific symptoms of dyspnea, lethargy and reduced appetite were more common among older patients than younger patients when all sites of TB were included, but only dyspnea was statistically significant [27]. The distribution of pulmonary and extrapulmonary disease is similar between the elderly and younger patients.

Comorbidities, which are more common in older patients, may mask the symptoms of TB. For example, those with chronic coughing due to COPD may have a delayed presentation or diagnosis. However, conversely, they may also have closer healthcare contact [27]. 

The commonest differential diagnosis in the elderly is bronchogenic carcinoma. Some authors showed that approximately 5% of patients, all elderly, presented with both a lung tumor and pulmonary tuberculosis. Other differential diagnoses include fibrotic lung disease, lymphomas and pneumonias due to other organisms [24]. Diagnosis in the elderly is delayed more often [28].

### 4.2. Radiological Findings

In pulmonary TB, differences are found in radiological findings between younger and elderly patients. Cavitation seems to be more common among younger than older patients [27,29]. In a cross-sectional study, in which elderly people were defined by age >60 years, bilateral multiple zone involvements were commonly observed in both the age groups. In addition, if infiltrates are the most common radiological finding whatever the age, they appear to be more frequent in elderly patients compared to younger (65.76% vs. 35.25%, *p* = 0.05) [30]. 

High resolution computed tomography (CT) is used to characterize active pulmonary disease where the standard chest radiograph is unhelpful and for extra-pulmonary disease [26]. Chest CT is very useful to overcome the limitations of chest radiography, such as poorer inspiration, associated pathologies, deformations of the chest wall [31]. CT can be useful for assessing differential diagnoses in symptomatic patients. In elderly patients, more precise criteria are needed to distinguish active tuberculosis from sequelae. CT can also help for organ lesions, sampling for tuberculosis culture using percutaneous biopsy in doubtful situations [32]. Early diagnosis of miliary TB relies heavily on CT findings [33].

The elderly patient may present with atypical radiological features, such as middle or lower lobe (rather than upper lobe) infiltrates, mass-like lesions or nodules appearing more like cancers, extensive bronchopneumonia without cavitation or nonresolving infiltrates. Lesions are frequently misdiagnosed as pneumonia or lung cancer in the elderly [34].

Fujimoto et al. investigated the value of diagnostic thoracoscopy in patients over 75 years of age with pleural effusion. The positive rate of culture of *M. tuberculosis* in pleural effusion without thoracoscopy was 24.4% (21/86) in patients over 75 years old in one hospital in Japan from January 2008 to December 2018. With thoracoscopy, the positive rate of culture of *M. tuberculosis* is 55.6% (10/18). Therefore, thoracoscopy under local anesthesia has proven to be more effective for the diagnosis of pleural effusion [35]. Laparoscopy is the diagnostic tool of choice for abdominal tuberculosis and allows inspecting the peritoneum as well as obtaining biopsies for microbiological and histological examination [36].

### 4.3. Microbiological Findings

The elderly are frequently unable to spontaneously produce sputum. Other methods to obtain specimens should be tried, such as bronchoscopy or induced sputum production by nebulized hypertonic saline [24]. 

In frail elderly patients, however, the risk of such a procedure should be carefully weighed against the benefit of potentially making a diagnosis of tuberculosis. With the potential risk of MDR TB, the importance of culturing all specimens for tuberculosis to identify the bacterium and to provide sensitivity testing is important. 

The sensitivity of microscopic examination of acid-fast bacilli (AFB) in sputum specimens is 50% or less, and almost always negative in infected pleural fluid. AFB staining is still the most widely used rapid diagnostic method for tuberculosis [37]. However, its value for patients who cannot produce sputum spontaneously is very little [38]. Miliary TB is usually classified as pauci-bacillary TB, despite the high mycobacterial antigenic burden. Only 30% to 65% of cases of miliary TB are positively diagnosed on the basis of sputum culture [39]. The samples are cultured in Lowenstein-Jensen (LJ) solid medium and in liquid medium after microscopic examination for acid-fast bacilli. The liquid culture medium has a 10 to 15% higher sensitivity than the LJ medium. The growing colonies are then studied for identification of the mycobacterium species. Susceptibility to four first-line drugs (rifampicin, isoniazid, streptomycin and ethambutol) is tested on positive culture and automated liquid antibiograms are very efficient [40]. The WHO recommends mycobacterial culture, which exhibits high sensitivity for detecting *M. tuberculosis* as the diagnostic gold standard. Unfortunately, due to the slow growth of *M. tuberculosis*, mycobacterial culture cannot meet the clinical needs of the diagnosis and treatment of TB.

Rapid molecular biology techniques complement traditional cultures, allowing rapid diagnosis and study of genotypic bacterial resistance particularly to rifampicin with the GeneXpert MTB/RIF assay [38]. Early diagnosis and analysis of drug resistance are crucial for effective patient management and prevention of the spread of MDR TB. Other techniques of molecular biology are being developed, such as the detection of circulating cell-free M. tuberculosis DNA (cfMTB-DNA), which has recently emerged as a tool for diagnosing pauci-bacillary forms of TB such as tuberculous meningitis and pleural TB [41,42]. 

More recently, metagenomic next-generation sequencing has been used in cases where the diagnosis of tuberculosis is difficult to make by the usual techniques, improving the detection of *M. tuberculosis* [43]. 

Culture of mycobacterium on biopsy tissues from various sites, such as the liver, lymph nodes, bone marrow, pleura and synovium, is necessary to diagnose extrapulmonary forms and can be helped by histological analysis that reveals the characteristic tissue reaction (caseous necrosis with granuloma formation) [44]. 

There are also indirect diagnostic tests such as the tuberculin skin test (TST) or Interferon gamma release assays (IGRAs), which are used to detect latent *M. tuberculosis* infection (LTBI) in adults that have been evaluated in older people [45,46,47,48,49]. TST positivity in a developed country tends to increase until approximately 65 years of age, after which the positivity of the tuberculin test decreases [50]. In addition, it is more difficult to perform on the fragile skin of elderly subjects [24]. The positive rate on TST was 55% in patients aged 70 to 79 years and 33% in patients >80 years. However, the positive rate on IGRA (QuantiFERONTB-2G®, QFT-2G) testing was 79% and 75%, respectively, and the indeterminate result rate of QFT-2G increased with age. It was suggested that this may be due to comorbidities [47,49,51,52]. IGRAs could potentially be used in elderly patients in whom active TB is suspected but where all other investigations are inconclusive or not practical because of the patient’s general condition [47]. Among IGRAs, some studies show a better sensitivity of the T-SPOT.TB® test vs. QuantiFERON® in the context of LTBI in elderly patients [45,46,49]. Indeed, patients with negative T-SPOT.TB results were older (median age 60 years [IQR 34.0–65.0]; *p* < 0.01) when compared to patients with positive results. The cellular immune response induced by *M. tuberculosis*-specific antigens gradually weakens, leading to a downward trend in the IFN-γ concentration with increasing age [53]. Overall, a negative LTBI indirect diagnostic assay, whether a TST or IGRA, does not rule out latent TB disease and a fortiori active infection, or differentiate between old and new *M. tuberculosis* exposure.

Diagnosis is often made on suspicion, and a probabilistic TB treatment may be started before microbiological confirmation is obtained. Diagnostic difficulties in the elderly are common in many diseases, not solely TB. Problems, such as poor memory, deafness, blindness or partial sight, and impaired speech all contribute, often making an accurate history difficult. The patient, family and doctor may often attribute symptoms to “old age”. Comorbidities often further complicate matters, especially malignancy that may often coexist [54,55,56].

## 5. Treatment 

There are no WHO recommendations for a specific TB treatment in the elderly, probably because of limited clinical data concerning the treatment of TB in the elderly population and the heterogeneity of these data from different types of populations studied (from different income countries, threshold of age for elderly, presence or not of an associated HIV infection) [9,57,58].

### 5.1. Treatment Tolerance

Several studies report contrasting data about a poorer tolerance of TB drugs in elderly patients. Among an Indian cohort of TB, reported side effects were higher in elderly patients (63%) vs. younger patients (54%) [59]. Similarly, the elderly had a higher frequency of adverse drug reactions (18.5% vs. 40.7%) in a Korean cohort [34]. These results are inconsistent with some studies in high-income countries showing that adverse TB drug reactions were similar or higher in younger adult patients than in the elderly. For example, among nursing home residents in Arkansas, isoniazid-related liver toxicity occurred in only 4.5% of those over 80 years of age compared to 3.5% of those aged 50 to 64 years. Digestive intolerance was reported in 6.9% and 4.1% of the same population [60]. Moreover, the overall rate of side effects leading to discontinuation or change of TB treatment was lower in older patients (21%) compared to younger ones (24%) [61].

The influence of aging in the risk of hepatotoxicity during TB treatment is also unclear, and involves, besides age-related physiological changes, potential other comorbidities and treatments [61,62,63]. The risk of drug-induced hepatitis and other serious adverse events could increase with age due to less efficient elimination of drugs as a result of reduced renal and hepatic clearance [64].

Because pyrazinamide is most frequently responsible for liver damage [62,65,66], the American Thoracic Society, the Centers for Disease Control and Prevention and the Infectious Diseases Society of America assess that in the elderly with a moderate TB disease with low risk of drug resistance, the benefits of pyrazinamide use in the initial dosing regimen is less than the risk of serious adverse events [62,67]. Therefore, American guidelines do not recommend the use of pyrazinamide during the intensive phase in patients aged >75 years for moderate disease and low resistance risk [68,69]. In this setting, the initial regimen consists of isoniazid, rifampicin and ethambutol.

The total duration of treatment for tuberculosis without pyrazinamide during the intensive phase should be extended to at least 9 months. When an elderly patient has active TB with a high bacillary load (cavitary forms), the addition of a fourth antibiotic is necessary to prevent the development of resistance, and the use of pyrazinamide or a fluoroquinolone (levofloxacin, moxifloxacin) may be considered [69]. Nevertheless, a recent study shows that in very elderly patients (>80 years) the use of pyrazinamide with careful monitoring can be safe and well tolerated [70].

### 5.2. Pharmacokinetic Issues and Drug-to-Drug Interaction

Frequent impairment of renal function complicates the use of TB drugs and warrants dose adjustment. For clearances between 30 and 50 no dose adjustment is necessary but regular dosing is warranted. For creatinine clearances <30 mL/min, it is recommended to reduce the daily dosage or space out the doses according to each molecule. The 2016 IDSA guidelines offer a dose adjustment chart [69]. Rifampicin and isoniazid are metabolized by the liver and conventional dosing can be used in renal failure. Pyrazinamide is metabolized by the liver, but its metabolites may accumulate in patients with renal impairment warranting a 48 h interval or three intakes per week for pyrazinamid and ethambutol [71,72]. 

Drug-drug interactions can alter the concentrations of the drugs involved. This problem should be considered in the elderly, who are often exposed to polymedication.

Few interactions significantly modify the concentrations of anti-tuberculosis drugs other than fluoroquinolones. The concomitant use of vitamin supplements containing calcium, iron and zinc and antacids reduces the absorption of fluoroquinolones. It is necessary to space out the intake of these drugs by at least two hours from that of the fluoroquinolones.

On the other hand, TB treatments modify the concentrations of other drugs. Rifampicin is an enzyme inducer. By inducing the activity of metabolic enzymes, rifampicin decreases the serum concentrations of many drugs, sometimes to subtherapeutic levels. 

The drugs frequently used in the elderly, and justifying monitoring with adaptation of doses or search for alternative drugs, includes in a nonexhaustive list: Benzodiazepines, Simvastatin, luvastatin, Verapamil, Nifedipine, diltiazem, Enalapril, Losartan, Glimepiride, Repaglinide, Propranolol, Metoprolol, Corticosteroids, Warfarin, and Levothyroxine.

Isoniazid is a relatively potent inhibitor of several isozymes. Isoniazid increases the concentrations of some neurotropic drugs used in the elderly such as carbamazepine and certain benzodiazepines and serotonergic antidepressants. This effect is offset by the inducing effect of rifampicin. This should be taken into account when discontinuing either drug [71].

Measuring plasma TB drugs concentrations has not been shown to be of therapeutic benefit but may be performed in cases of suspected noncompliance and suspected drug interactions to guide dosage adjustment [69]. 

## 6. Outcome

### 6.1. Specific Data in Elderly People

The elderly population is schematically divided into three types of subjects according to their health status [73]: vigorous elderly people in good health who are independent and autonomous; frail elderly people who are distinguished by a lessening of their ability to deal with stress, no matter how small, and dependent elderly people in poor health due to chronic polypathology causing handicaps. Frail or dependent elderly people are not only at greater risk of contracting an infection than vigorous elderly people, but are also at greater risk of presenting serious complications in the case of infection. Recognising frailty to prevent loss of independence is a major challenge in dealing with the therapeutic aspects of elderly patients. The most relevant clinical markers of a state of frailty are malnourishment, falls, incontinence, cognitive disorders (Alzheimer’s disease or similar, and delirium).

The management of an elderly TB patient is, therefore, ideally based on an adaptation of the treatment based on a standardized gerontological assessment [74]. Frailty screening tools exist, allowing clinicians to refer the patient for geriatric evaluation or not, as cancerologists do with the G8 tool. These tests, for example the Edmonton Scale, most often assess cognition, general condition, functional addictions, social support, multiple medication, and risk for falls [75].

The treatment of TB in the elderly requires adaptations linked to the specific nature of aging. Regardless of the role of ethambutol, which is questionable in the elderly, tolerance of the treatment is often poor and requires therapeutic adaptations, which are sometimes difficult. In addition, the elderly are often cognitively impaired, socially isolated, polypathological and fragile. They are therefore exposed to a major risk of poor compliance, drug interaction and intolerance to treatments, in particular linked to renal failure and hypoalbuminemia.

Adherence and tolerance of TB treatment involve education and training of both caregivers and patients and, if necessary, the attention of the health professional on a daily basis to ensure the dispensing of drugs. Adherence to TB treatment is a fundamental issue in the outcome if lack of strict adherence to treatment exposes the patient to the risk of therapeutic failure and the emergence of anti-tuberculosis drug resistant strains. The actual intake is easy to control, due to the expected side effects of the treatment such as red urine (due to rifampicin) or hyperuricemia (due to pyrazinamide). Nevertheless, taking the daily treatment on an empty stomach, to improve oral bioavailability of anti-TB drugs (rifampicin, isoniazid), can pose nutritional problems, since breakfast is important for the elderly. Older patients with active TB are frequently undernourished because of the infection, so physicians need to be particularly attentive to obtain good nutritional status for patients with the support of a dietician, use of food supplements and vitamin supplementation. Although there is little evidence for routine nutritional and vitamin supplementation [76], daily administration of pyridoxine (vitamin B6) during isoniazid treatment is recommended, including in elderly patients [69,77]. Adherence support aimed at ensuring successful tuberculosis treatment has historically relied on the use of directly observed therapy (DOT) [78]. The use of DOT has shown mixed results in multiple studies and metaanalyses, largely because the term appears to be a catchall phrase for different treatment support approaches [79]. When coupled with emotional support, nutritional supplementation, and other types of enablers, DOT can be a way to ensure daily contact with vulnerable individuals and close monitoring for the development of adverse events. DOT is, therefore, particularly well suited to frail elderly subjects. DOT performed by the family is, nevertheless, inferior to DOT performed by a provider [77,78,79].

Recently studies have been shown the superiority of innovative strategies based on new technology, such as virtually observed therapy using smartphones [80]. However, these strategies were conducted in young people, and are probably not enforceable with older people who have little access to new technologies such as smartphones. 

In addition, it is recommended that fixed combinations be used in preference, as they simplify the taking of medication and compliance [2].

### 6.2. Paradoxical Reaction during TB Treatment 

Many other infectious diseases with local and systemic symptoms of TB are caused by the growth of *M. tuberculosis* and the host’s inflammatory response to the presence of the bacteria in the tissue [81]. TB infection initiates an inflammatory immune response that causes tissue damage. In addition, active TB infection by itself appears to be an immunosuppression factor, via the modification of the immune response, which favors the long-term persistence of the bacteria in the tissue [82,83]. Hence, during adequate sterilizing antimycobacterial treatment, immunopathological reactions, due to gradual restoration of pathogen-specific immune responses, may occur with paradoxical worsening and upgrading clinical reactions [84]. However, these paradoxical reactions during TB treatment occurred less frequently, and with less exaggeration, than immune reconstitution inflammatory syndrome with dynamics of rapid immune restoration in immunocompromised individuals [85], such as HIV-infected subjects on antiretroviral therapy [15]. These paradoxical clinical worsening in non-HIV-patients, generally self-limited, are more common in lymph node TB, consistent with exacerbation of pain and swelling following the initiation of chemotherapy [85]. In addition, age was not found to predict the occurrence of paradoxical upgrading reactions in a retrospective analysis [86]. Adjunctive transient corticosteroid therapy may be used to treat paradoxical upgrading reactions. Frequent complications associated with corticosteroid therapy among the elderly (osteoporosis with osteoporotic fractures, falls or osteoarticular problems, protein-energy malnutrition, amyotrophy, and psychiatric complications) should be prevented and treated because they may have serious consequences in this frail population [87]. Attention should be paid to the prescription of preventive measures through comprehensive care.

## 7. Conclusions

Tuberculosis in the elderly is not a rare infection and requires special management. If the immunosenescence mainly favors the reactivation of the infection, whether or not aggravated by potential associated treatments (corticosteroids, immunosuppressants, anticancer chemotherapy), the search for HIV infection must be systematic. The average age of discovery of HIV is progressing by year after year, specially in high-income countries. The treatment of TB in the elderly is complex, combining the initial constraints of respiratory isolation and contact screening, often in long-term care facilities, and a prolonged use of combinations of anti-tuberculosis drugs that are potentially toxic and induce drug-drug interactions, in the context of often precarious general condition (undernutrition, co-morbidities, cognitive disorders) due both to active systemic infection and to old age. Multidisciplinary management, associating geriatricians and infectious disease specialists, based on close collaboration is justified throughout care to optimize a favorable outcome in these vulnerable patients.

## Figures and Tables

**Figure 1 jcm-10-05888-f001:**
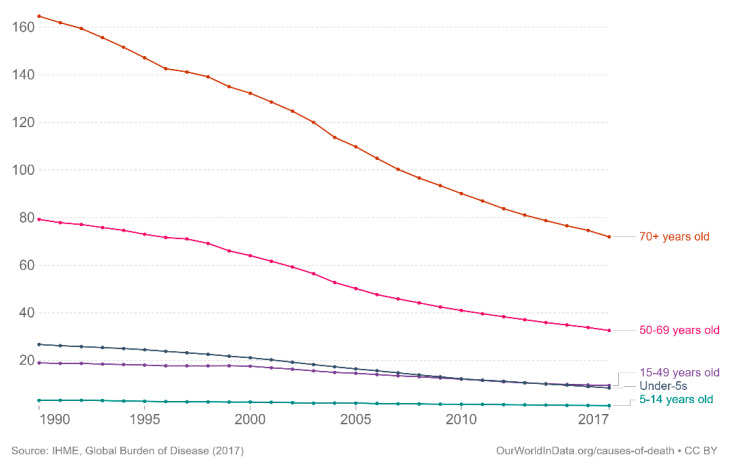
Death rate from tuberculosis by age in the World, 1990 to 2017 [8].

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
