# Peer review of "Tuberculosis in the Elderly"

_jcm, 2021, doi:10.3390/jcm10245888_

Round 1

Reviewer 1 Report

The authors have incorporated the modifications and the article can be processed for proof reading.

Reviewer 2 Report

In line 241 M.tuberculosis must be in italics

This manuscript is a resubmission of an earlier submission. The following is a list of the peer review reports and author responses from that submission.

Round 1

Reviewer 1 Report

This is a well drafted manuscript. I applaud the authors for meticulously compiling the already existing data and presenting it to the readers. The authors have covered every probable aspect to discuss senescence and tuberculosis, starting from epidemiology, pathophysiology, diagnosis, and treatment. 

I donot have any concern with the article except one.

Line 433: please remove the 3 dots after psychiatric complications.

Reviewer 2 Report

The authors provided a comprehensive review of TB in the elderly. Considering the increasing proportion of elderly TB, this manuscript is timely. However, I would like to raise some concerns as follows.

  1. epidemiology

(1) Line 44-58: This is too long. This manuscript is about elderly TB. Please focus more on TB and age.

(1) Line 59-63: please show the reference regarding the aging TB epidemic in some regions and consider a figure showing it.

(2) Line 69-72: too decisive expressions

(3) Line 73: What is the difference between 2.1 TB outcome and 6. outcome? You may streamline the structure.

(4) Figure: There is no explanation about the rate (per 100 000?).

(5) LIne 101-103: The statement is not evidence-based. Please provide the relationship between the proportion of reactivation and the risk of DR-TB.

(6) Line 105-121: This part is focusing only on the relationship between HIV and aging. There is no clear statement about the risk of TB among the elderly with HIV. better to exclude if no significant findings.

2. Diagnosis

(1) Line 225-254: The explanation about the test is too long and there is no age-related statement, for example, the sensitivity of the test is lower among the elderly. better to exclude or simplify if no significant findings.

(2) Line 255-274: IGRA and TST are not microbiological tests, but immunologic tests. better to rearrange.

(3) Line 276-282: better to move to clinical findings.

3. Outcome

(1) Line 383-411: Why is this part included in the outcome section? better to rearrange, for example, treatment adherence in the treatment section.

(2) Line 413-435: Why is this part included in the outcome section? This happens during the treatment. seems like no need to make the outcome section.

Reviewer 3 Report

This is an elegantly written review on tuberculosis on elder patients discussing the clinical information and outcome in the context of epidemiology, pathophysiology and diagnosis. Although authors have covered most of the part, here is a couple of thoughts to strengthen the article.

  1. Authors are recommended to include a section on biomarkers. Whole bunch of prognostic information is available in the literature
  2. Adding up a schematic representation always add more weightage to the review article. Authors could consider that for the diagnosis section.
  3. Authors may add another section providing their perspective/opinion in the context of unmet medical needs and also strengthen the conclusion section.